# Is Europe prepared to go digital? making the case for developing digital capacity: An exploratory analysis of Eurostat survey data

**Robin van Kessel**[1,2,3☯]*, **Brian Li Han Wong**[3,4,5,6☯], **Ivan Rubinić**[2], **Ella O'Nuallain**[7,8], **Katarzyna Czabanowska**[1,9]

**1** Department of International Health, Care and Public Health Research Institute, Maastricht University, Maastricht, Netherlands, **2** Studio Europa, Maastricht University, Maastricht, Netherlands, **3** Research Committee, Global Health Workforce Network (GHWN) Youth Hub, World Health Organization (WHO), Geneva, Switzerland, **4** Medical Research Council Unit for Lifelong Health and Ageing at UCL, Department of Population Science and Experimental Medicine, UCL Institute of Cardiovascular Science, University College London, London, United Kingdom, **5** Steering Committee, Digital Health Section, European Public Health Association (EUPHA), Utrecht, Netherlands, **6** COVID-19 Task Force, Association of Schools of Public Health in the European Region (ASPHER), Brussels, Belgium, **7** Consulting team APAC Region, Global Hub, Impact Consulting, London, United Kingdom, **8** Public Sector Strategy team, Consulting, Deloitte Consulting Pty Ltd, Sydney, Australia, **9** Department of Health Policy Management, Institute of Public Health, Faculty of Health Care, Jagiellonian University, Krakow, Poland

☯ These authors contributed equally to this work.
* r.vankessel@maastrichtuniversity.nl

**Data Availability Statement:** The data used in this study and the data dictionary are openly available through the Eurostat portal (https://ec.europa.eu/

## Abstract

Digital divides are globally recognised as a wicked problem that threatens to become the new face of inequality. They are formed by discrepancies in Internet access, digital skills, and tangible outcomes (e.g. health, economic) between populations. Previous studies indicate that Europe has an average Internet access rate of 90%, yet rarely specify for different demographics and do not report on the presence of digital skills. This exploratory analysis used the 2019 community survey on ICT usage in households and by individuals from Eurostat, which is a sample of 147,531 households and 197,631 individuals aged 16-74. The cross-country comparative analysis includes EEA and Switzerland. Data were collected between January and August 2019 and analysed between April and May 2021. Large differences in Internet access were observed (75-98%), especially between North-Western (94-98%) and South-Eastern Europe (75-87%). Young populations, high education levels, employment, and living in an urban environment appear to positively influence the development of higher digital skills. The cross-country analysis exhibits a positive correlation between high capital stock and income/earnings, and the digital skills development while showing that the internet-access price bears marginal influence over digital literacy levels. The findings suggest Europe is currently unable to host a sustainable digital society without exacerbating cross-country inequalities due to substantial differences in internet access and digital literacy. Investment in building digital capacity in the general population should be the primary objective of European countries to ensure they can benefit optimally, equitably, and sustainably from the advancements of the Digital Era.

eurostat/web/digital-economy-and-society/data/
database).

**Funding:** The authors received no specific funding
for this work.

**Competing interests:** The authors have declared
that no competing interests exist.

## Introduction

The digitalisation of society is a process that has not only been occurring over the past seven decades, but has been significantly accelerated by the COVID-19 pandemic. Digitalisation is one of three fundamental processes propelling change in the economic, socio-political, and cultural spheres globally, alongside globalisation and demographic change [1]. Work, social interactions, health and education services, and recreational activities have been all forced to (further) adopt digital technologies and platforms, as most of the world transitioned to digital environments at the beginning of the COVID-19 pandemic [2]. However, not only is more attention drawn to digital divides through COVID-19, digital and technological advances have neither equally nor equitably permeated all layers of society across the globe, further widening existing digital divides [3–5]. As such, digital divides are a global wicked problem and threaten to become the new face of inequality due to their potentially detrimental effects on the Sustainable Development Agenda [6,7].

In its 2016 Skills Agenda for Europe, the European Commission first recognised the great necessity for digital skills development [8] as – at the time of writing – almost half of the European Union's (EU) population lacked basic digital skills, while 20% had none at all. This is even more problematic considering the rapid progression into the Digital Age due to the COVID-19 pandemic—specifically the shifts in healthcare and public health [9–12], but also to wider determinants in health, such as ways of working and access to education [3,10,13,14]. One example is the use of AI-assisted diagnoses and the provision of telehealth for physical and mental health issues, driven by the symbiotic need to keep patients out of hospitals where they might have a high risk of spreading or contracting covid and limiting physical contact with medical staff [15]. Another example is the prominent role that social media has played in spreading factually incorrect information during the pandemic [16–18]. Digital skills can no longer be considered luxuries; they are foundational to modern governance, societal and economic functioning, and access to parts of the healthcare and public health systems [14]—as reflected by the European Skills Agenda 2020 update [19].

While COVID-19 has showcased that it is possible to live in a digital society, it exacerbated existing health and economic inequalities and widened digital divides [6]. The question thus remains whether the current European ecosystem is fit – let alone adequately prepared – to safely and sustainably host a digital society. To illustrate, the United Kingdom's Office for National Statistics showed in 2019 that 10% of the adult population did not use the Internet and Global Kids Online reported in 2021 that 14% of children under 19 in Europe still did not have frequent internet access [20,21]. However, these statistics only reflect the first level of digital divides (binary Internet access), while the second (internet skills) and third levels (tangible and beneficial outcomes of internet use) remain largely unexplored. Demographic characteristics, which are a known influence on the development of digital skills [3,22], have sparsely been accounted for in previous research.

This article, therefore, analyses the first two levels of digital divides among people aged 16-74 in Europe and explores the influences on the third level. Findings are interpreted through Greenhalgh's diffusion model to explore differences in uptake [23] and the digital determinants of health to explain health inequalities [22,24,25] (see Text A in S1 Text). Consequently, it is possible to understand the continued existence of digital divides and their societal effects, as well as propose multi-disciplinary recommendations for global governance.

## Methods

### The community survey and data access

The community survey on ICT usage in households and by individuals provides a set of variables that, when combined, measure an individual's level of digital skills. They are based on the

2016 Digital Competence Framework of the European Commission in which key skills relating to digital technologies are set out [8]. Proficiency in digital skills is divided into four levels: above basic digital skills, basic digital skills, low digital skills, and no digital skills. The level of digital skills is determined by the reported outcomes in four domains: information skills, communication skills, problem-solving skills, and software skills (for content manipulation). Further breakdown of the domains into their respective subcategories is provided in Table A in S1 Text.

Individuals with above basic digital skills score "above basic" in all four domains; individuals with basic digital skills score at least one "basic" but no "no skills" in all four domains; individuals with "low" digital skills (i.e. missing some type of basic skills) score one to three "no skills" in the four domains; and individuals with "no digital skills" score "no skills" in all four domains. In other words, they report having no activities performed in all four domains, despite indicating that they used the internet at least once during the last three months.

The community survey on information and communications technology usage in households and by individuals was completed across European households and individuals between January and August 2019 [26]. Households and individuals were sampled through quota sampling methods (n = 1; Germany), one-stage sampling techniques (n = 2; Malta and Lithuania), or probability sampling (n = 25; the remaining 24 EU Member States and the UK) [26]. The exact means through which households or individuals were recruited differed per EU Member State (see 'Methodological Manuals' [26]). Data were collected via face-to-face interviews, telephone interviews, and postal surveys. However, there is some degree of heterogeneity in the methods used across European countries. The overall accuracy of the survey data is considered high, as it represents approximately 75% of the population between the ages of 16 and 74 [27]. On average, the net sample size counted between 3,000 and 6,000 units per country, amounting to a total of 147,531 households and 197,631 individuals. Therefore, this study operates under the assumption that the findings of this study are an accurate reflection of the real situation among adults aged 16-74 in the European Union. Data was openly accessible through Eurostat. Further information about the methodology is provided in Text B in S1 Text.

## Variable selection and coding

In order to properly select, understand, and interpret our findings and to infer possible consequences, we first established a directed acyclic graph showing theoretical pathways of how demographic variables affect the first and second levels of digital divides (Fig 1) [3,5,6,12,13,22,28]. Fig 1 allows us to take a theory-driven approach to the data analysis by outlining how variables are related in causing digital divides. It, therefore, allows us to extract relevant variables from the community survey to be included in the exploratory analysis: Internet access, digital skills, age, education level, employment, income, and urban status. Sex was only accessible through data on education level. Data on health status was not available.

Digital skill levels are divided into "above basic", "basic", and "low." The category "no digital skills" was removed from the analysis due to the unavailability of data. Further explanation on the digital skills' classification is provided in Table A in S1 Text. Individuals were clustered into five age groups (16-24, 25-34, 35-44, 45-54, 55-64, and 65-74 years). Sex was coded binarily. Education level was divided into three groups (low, medium, and high). Education level was further stratified by sex and three age groups (16-24, 25-54, and 55-74 years; data for the five age groups mentioned earlier were unavailable, hence the adjusted groups). Urban status was categorised into three levels (urban, suburban or town, and rural) according to the Eurostat Reference and Management of Nomenclatures [29]. Employment status was categorised into four categories (employed, unemployed, retired, and student). Income was classified

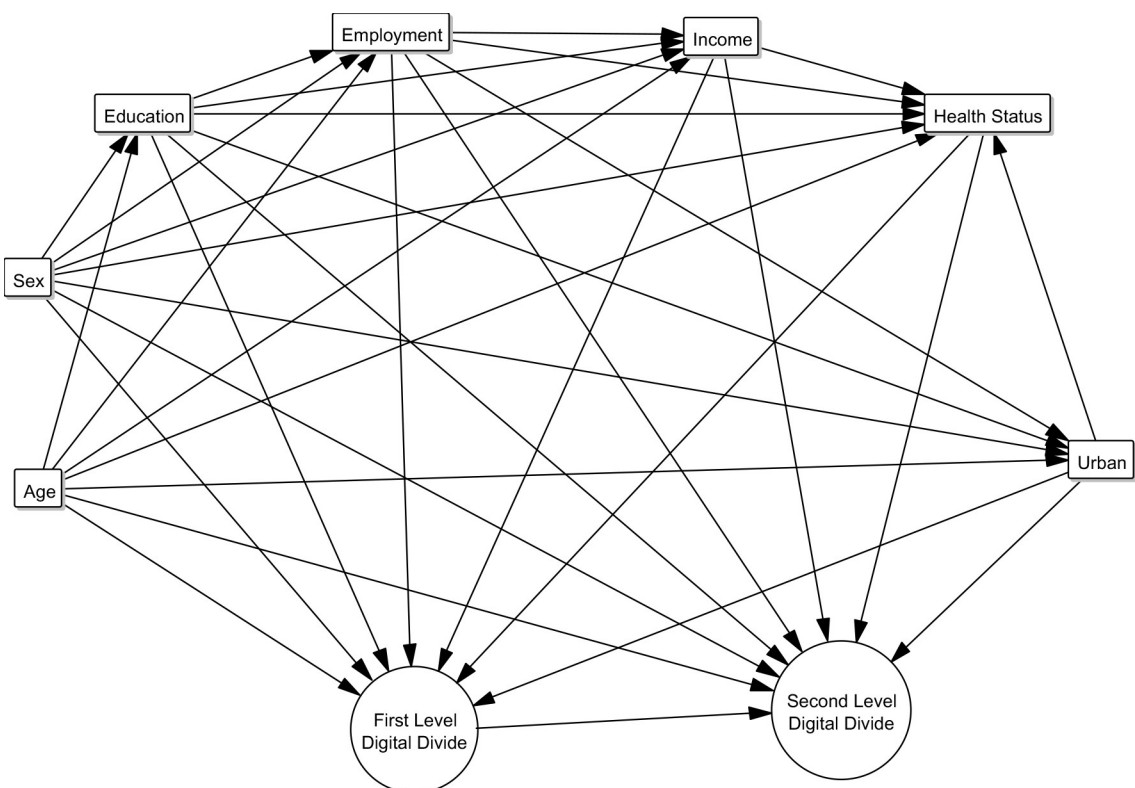

**Fig 1. A directed acyclic graph portraying the different causal pathways which influence internet access and digital skills development.**

through the distribution of national income by quartiles. All EEA Member States were included, in addition to Switzerland due to geographic, cultural, and developmental similarities with the EU Member States.

Internet access was binary-coded and households were grouped into four categories (all, urban, suburban or town, and rural). The types of devices used to access the Internet were divided into three categories (mobile devices, desktop or laptop, and both). It is important to note that the variable for the type of mobile device used to access the Internet was only available in the 2018 survey, hence that data was used.

## Data analysis

Data was cleaned in R version 4.0.2, specifically using the 'dplyr' and 'reshape2' packages. Further visualisation of data was performed in Tableau version 2021.1 to create scatter plots as well as both the geographical and traditional heatmaps. Descriptive data was examined in R using the 'summarytools' package. The exact code that was used with these packages is made openly available at https://github.com/robin-van-kessel/digitalskillsEurope/blob/8dae168c5b12fdec07431e1c11c125344bfd5e5e/Rcode.

## Results

### Descriptive statistics

Since internet access was based on households rather than individuals, the descriptive statistics are reported separately. Overall, Internet connectivity of 89.12%(SD: 6.07%) was reported.

Urban regions report 91.59%(4.43%) Internet access, whilst suburban or towns report 89.28% (6.04%), and rural areas report 84.97%(10.19%). Households in the first income quartile report an across-the-board-average Internet connectivity rate of 72.83%(15.01%), the second quartile 86.50%(9.31%), the third quartile 95.33%(3.89%), and the fourth quartile 98.86%(1.03%).

Overall, the highest proportion of people reported having above basic digital skills (34.65% [12.13%]), followed by low digital skills (27.48%[7.69%]), and basic digital skills (24.23% [4.47%]). Above basic digital skills were most common in the age group 16-24 (60.35%[15.36%]) and least in the age group 65-74 (7.87%[5.86%]). Individuals with high levels of education report higher levels of digital skills relative to medium and low levels. The distribution of the higher levels of digital skills across all educational levels is consistently skewed in favour of males. Individuals living in urban regions report higher levels of above basic digital skills (40.81%[12.40%]) compared to rural regions (28.87%[12.37%]). Students and employed people report the highest levels of above basic skills (68.16%[14.18%] and 40.66%[12.75%]). People with higher incomes also tend to have higher levels of above basic digital skills (48.77%[14.99%]) than people with lower incomes (20.15%[12.57%]). What is particularly striking is the large spread of data (indicated by the large standard deviations), which can be attributed partially to the heterogeneity in how data was collected (a mix of households and individuals and sampling methods), but also highlights that it may be ill-advised to interpret the European Union as a homogenous region in terms of digital skills. All descriptive statistics are presented in Table 1.

## Internet access across Europe

Internet access is accessible to the majority of the European countries' populations (ranging from 75 to 98%). Nevertheless, there are certain geographical discrepancies between different regions. The North and North-West are shown to have noticeably higher internet access rates compared to other European regions (94-98% compared to 89-91% in Central and South-West and 75-87% in South and East). Urban regions show more consistent levels of internet connectivity than rural regions (82-99% versus 62-99%). Notable differences are found in rural Southern and Eastern Europe (62-75%) compared to urban (82-87%). Low income may also be associated with lower levels of internet access (33-96% compared to 65-100%; 83-100%; and 96-100% for the second, third, and fourth income quartile respectively). Fig 2 shows the overall geographic distribution of household internet access in Europe, stratified by urban status and income level. Further explanation on what devices are used to access the Internet is included in the Text C and Table B in S1 Text.

## Digital skills across Europe

The majority of highly digitally skilled people are found in the Northern and North-Western parts of Europe, indicating that over 50% of individuals possess above basic digital skills. In contrast, South-Eastern Europe shows less than 20% of individuals having above basic digital skills. This is reversed in individuals with low digital skills, who are frequently reported in South-Eastern Europe, while less reported in North-Western Europe. Fig 3 shows the country-level distribution of digital skills in Europe.

Based on the spread of data, low age can be associated with more developed digital skills, whereas high age tends to be associated with lower digital skills. That being said, the development of above basic digital skills among younger people is inconsistent in Europe as indicated by the reported range (22-85%). Looking back at Fig 1, it is safe to assume that age needs to interact with other factors to reliably produce high digital skills. The presence of above basic digital skills tends to decline with age, whereas low digital skills increase with age. Above basic digital skills are mostly reported in high education (29-84%), while least in low education (2-

**Table 1. Descriptive Statistics.**

| | | | Digital Skills | | | | | |
| --- | --- | --- | --- | --- | --- | --- | --- | --- |
| | | | Above Basic Digital Skills | | Basic Digital Skills | | Low Digital Skills | |
| | | | Mean (%) | SD (%) | Mean (%) | SD (%) | Mean (%) | SD (%) |
| All Individuals | | | 34.65 | 12.13 | 24.23 | 4.47 | 27.48 | 7.69 |
| Age Group | 16-24 | | 60.35 | 15.36 | 23.03 | 7.70 | 14.84 | 9.30 |
| | 25-34 | | 50.65 | 14.93 | 25.52 | 6.09 | 21.00 | 11.14 |
| | 35-44 | | 41.61 | 14.05 | 27.52 | 5.27 | 26.48 | 10.79 |
| | 45-54 | | 30.68 | 13.55 | 26.77 | 5.94 | 32.48 | 10.79 |
| | 55-64 | | 18.81 | 10.50 | 23.68 | 8.49 | 35.32 | 7.65 |
| | 65-74 | | 7.87 | 5.86 | 17.68 | 11.27 | 31.94 | 7.95 |
| Level of Education | High | | 57.03 | 11.60 | 27.29 | 5.50 | 13.19 | 7.02 |
| | | Female | 53.90 | 11.31 | 29.06 | 5.74 | 14.48 | 7.01 |
| | | Male | 60.48 | 12.55 | 25.19 | 5.95 | 11.84 | 7.23 |
| | | 16-24 | 74.72 | 15.14 | 18.04 | 9.76 | 6.12 | 7.18 |
| | | 25-54 | 63.29 | 11.90 | 25.35 | 6.75 | 10.23 | 6.66 |
| | | 55-74 | 33.65 | 11.80 | 34.58 | 6.66 | 23.84 | 9.40 |
| | Medium | | 29.68 | 12.87 | 26.81 | 6.09 | 32.16 | 10.01 |
| | | Female | 27.68 | 11.63 | 27.13 | 6.85 | 33.77 | 9.70 |
| | | Male | 31.58 | 14.03 | 26.55 | 5.74 | 30.74 | 10.68 |
| | | 16-24 | 62.03 | 15.02 | 22.74 | 7.36 | 14.32 | 9.67 |
| | | 25-54 | 31.19 | 14.23 | 30.48 | 6.46 | 32.97 | 14.25 |
| | | 55-74 | 12.16 | 8.11 | 22.68 | 11.01 | 39.39 | 7.56 |
| | Low | | 20.39 | 12.96 | 16.61 | 6.48 | 33.42 | 8.76 |
| | | Female | 18.16 | 12.49 | 15.58 | 7.50 | 33.71 | 10.39 |
| | | Male | 23.19 | 13.64 | 17.77 | 6.00 | 33.06 | 8.27 |
| | | 16-24 | 51.93 | 18.92 | 26.28 | 12.05 | 19.03 | 10.62 |
| | | 25-54 | 13.61 | 12.43 | 18.87 | 9.34 | 46.58 | 11.10 |
| | | 55-74 | 3.81 | 4.49 | 10.48 | 10.64 | 31.87 | 13.45 |
| Urban Status | Urban | | 40.81 | 12.40 | 24.42 | 3.91 | 24.35 | 7.90 |
| | Suburban or Town | | 33.03 | 12.41 | 24.45 | 4.97 | 28.94 | 8.42 |
| | Rural | | 28.87 | 12.37 | 23.61 | 6.90 | 29.42 | 8.63 |
| Employment Status | Employed | | 40.66 | 12.75 | 27.31 | 4.48 | 26.25 | 9.85 |
| | Unemployed | | 25.28 | 15.24 | 22.90 | 8.32 | 34.14 | 10.47 |
| | Retired | | 11.09 | 9.22 | 19.16 | 9.58 | 35.31 | 6.99 |
| | Student | | 68.16 | 14.18 | 20.59 | 8.83 | 9.94 | 7.63 |
| Income Level | First Quartile | | 20.15 | 12.57 | 18.04 | 6.94 | 31.77 | 7.39 |
| | Second Quartile | | 24.42 | 11.23 | 23.85 | 6.49 | 32.27 | 7.78 |
| | Third Quartile | | 34.35 | 12.83 | 27.35 | 5.28 | 28.42 | 9.84 |
| | Fourth Quartile | | 48.77 | 14.99 | 27.19 | 4.72 | 19.27 | 10.22 |

48%). The opposite applies to low digital skills (2-29% and 20-51% respectively). Urban areas show slightly more consistent above basic skills (17-63%) compared to rural areas (5-61%), while rural areas report a similar spread of low digital skills (13-46%) as urban areas (11-43%). Students and employed people indicate having the highest proportion of above basic digital skills (29-87% and 13-66% respectively). Finally, high income is associated with the development of higher digital skills (17-74%), although we have to note the wide spread here as well, implying that higher-income levels alone are not sufficient. Further details on the stratified distribution of digital skills per country are shown in Fig 2.

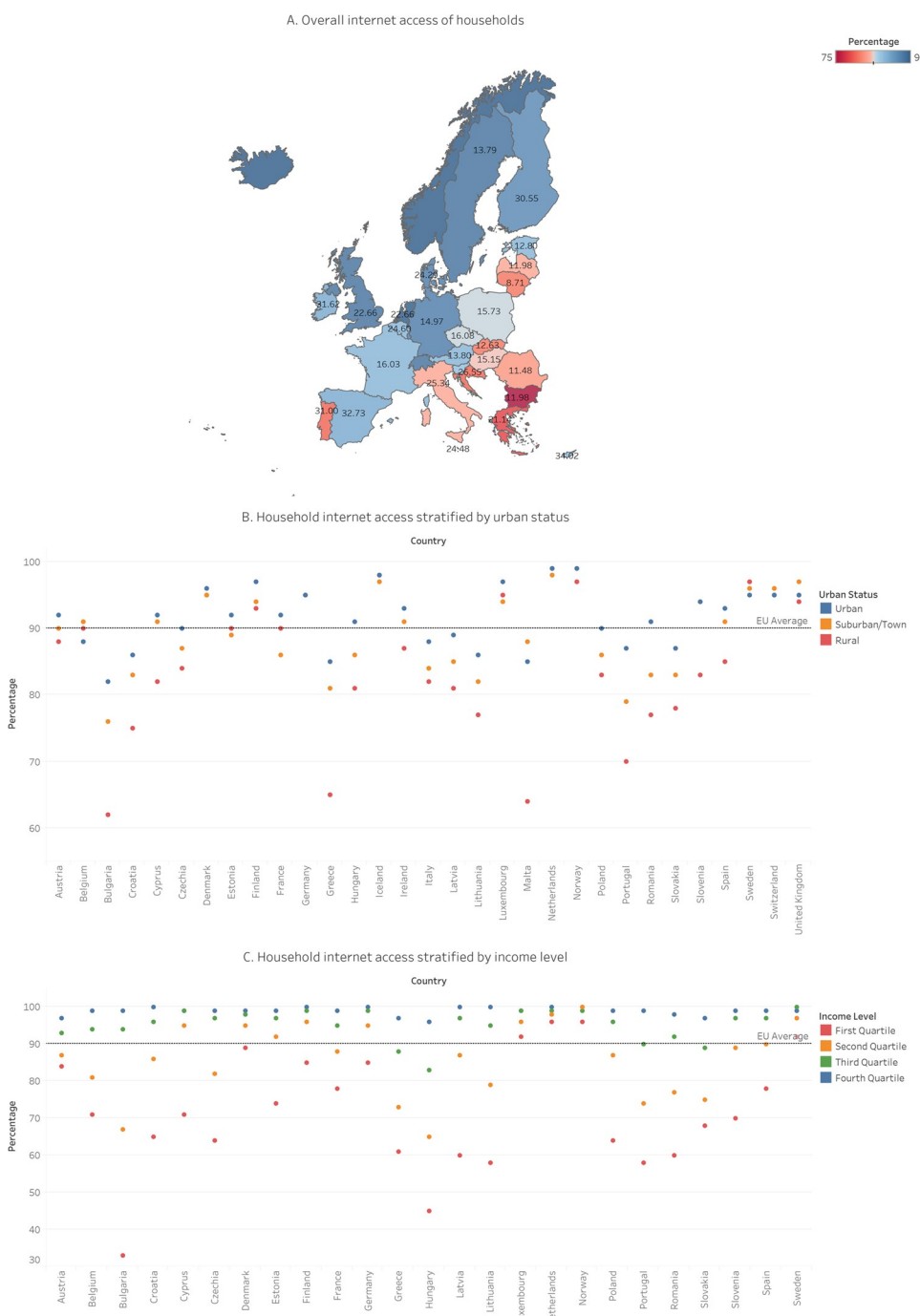

**Fig 2.** The extent to which European countries have access to the Internet overall and the normalized prices of standalone internet (12-30MB/s, in EUR PPP) (Fig 2A) [30], stratified by urban status (Fig 2B), and by household income level (Fig 2C). The maps were generated in Tableau using OpenStreetMap data (OpenStreetMap Contributors).

Figs 2 and 3 indicate that there exists a positive cross-country correlation between the high capital stock and income/earnings and the digital skills development. Moreover, the comparative analysis shows that cost of internet access bears marginal influence over digital literacy levels. These findings are consistent with the affordability of internet access prices, i.e., with their

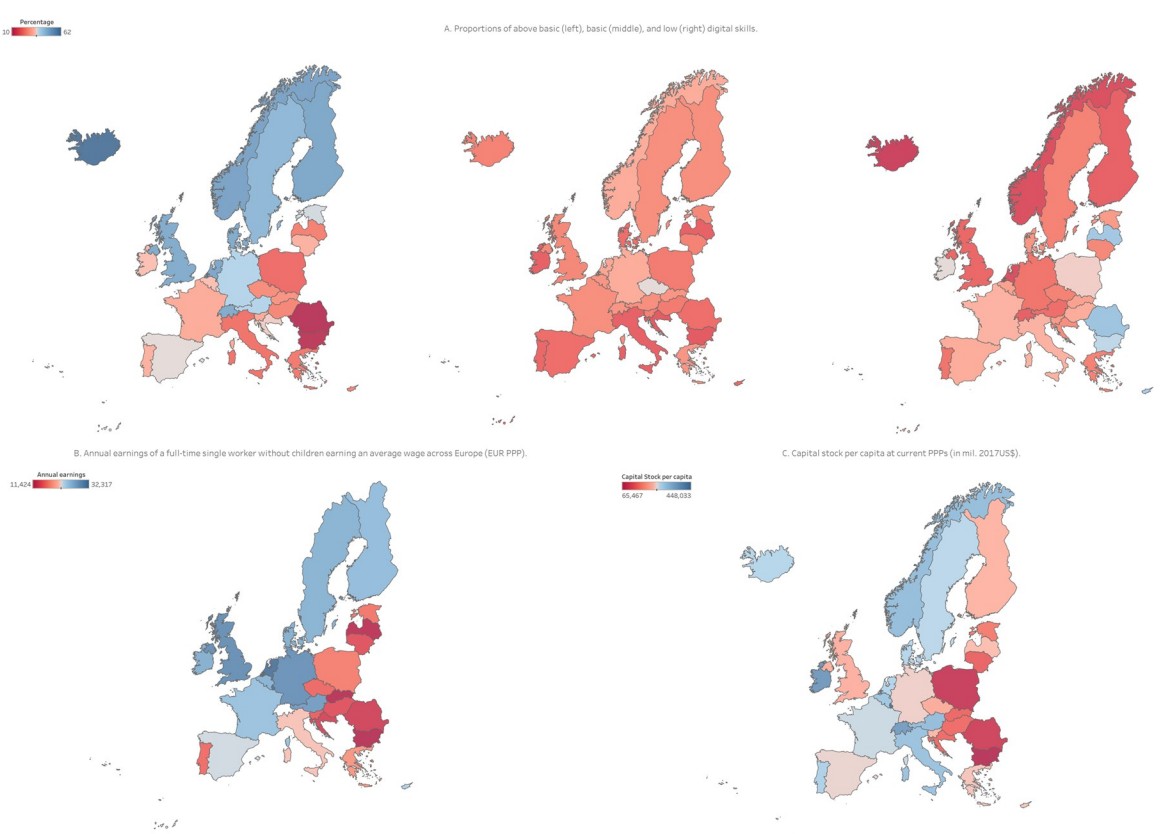

**Fig 3.** A geographical heat map of the distribution of digital skills (Fig 3A; left shows above basic skills, centre shows basic digital skills, and right shows low digital skills), annual net earnings (Fig 3B) [31], and capital stock per capita (Fig 3C) [32]. The maps were generated in Tableau using OpenStreetMap data (OpenStreetMap Contributors).

small share within the average income/earnings. The latter is to say that the influence of the European cross-country capital/income/earnings stratification concerning the digital skills levels is complex and cannot be reduced to internet service pricing alone. The decline of above basic and low digital skills is also observed in education levels, urban status, and income level and – along with the age-related decline – are shown per European country in Fig 4.

Educational stratification shows large differences in above basic digital skills in the age category 16-24 (43-96% in high education versus 16-86% in low education). The age groups 25-54 and 55-74 with high education exhibit higher levels of above basic skills (34-89% and 13-63% respectively) compared to medium and low education (5-55% and 0-46%; and 0-28% and 0-15% respectively). Females with higher education report lower levels of above basic digital skills (29-82%) compared to males (28-87%). For medium and low levels of education, males report above basic skills (6-56% and 3-55%) more frequently than females 6-51% and 1-47%). Low digital skills are underreported in males with medium and low education (14-57% and 18-50%) relative to females (17-53% and 18-57%), as shown in Fig 5.

## Discussion

The main purpose of this article is to examine the magnitude of the first two levels of digital divides within Europe: internet access and digital skills. The findings of this article highlight that Europe cannot be considered a monolith when it comes to internet access or digital skills.

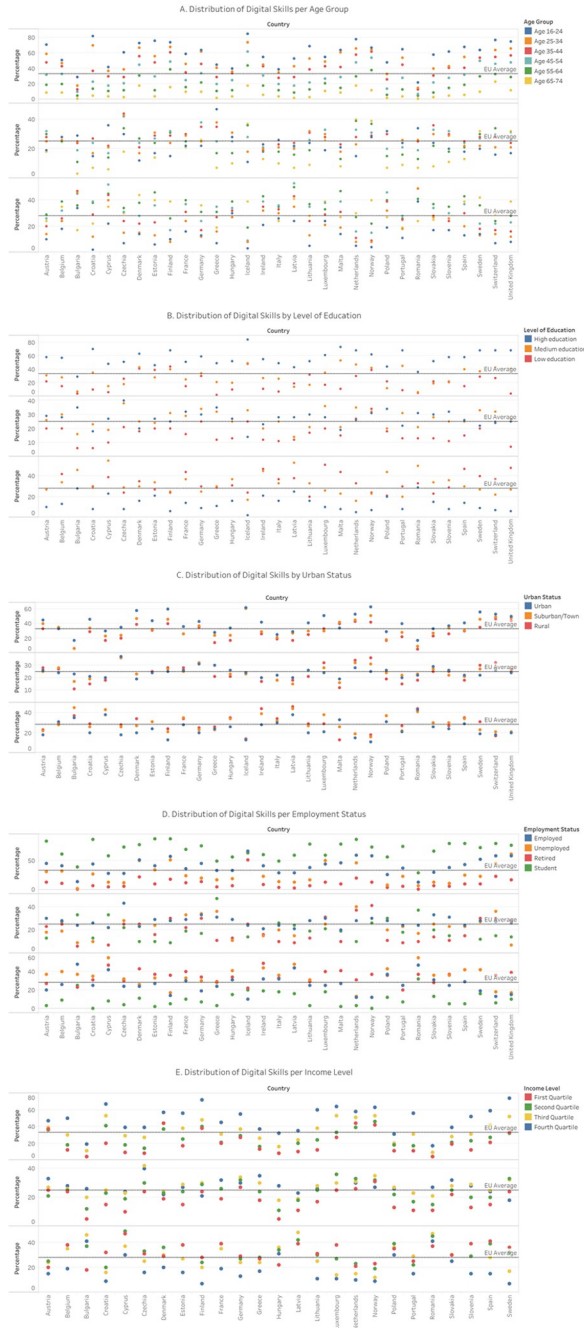

**Fig 4.** Scatterplot of digital skills across Europe stratified by age (Fig 4A), level of education (Fig 4B), urban status (Fig 4C), employment status (Fig 4D), and income level (Fig 4E). The top plot indicates above basic digital skills, the middle plot basic digital skills, and the bottom plot low digital skills.

In other words – while Europe may outperform other world regions on average in internet access and digital skills – substantial in- and cross-country inequalities still persist.

Overall, large differences are observed between North-Western Europe and the other regions, with internet access and digital skills being substantially higher in the former. When stratifying by age, level of education, sex, urban status, employment status, and income level,

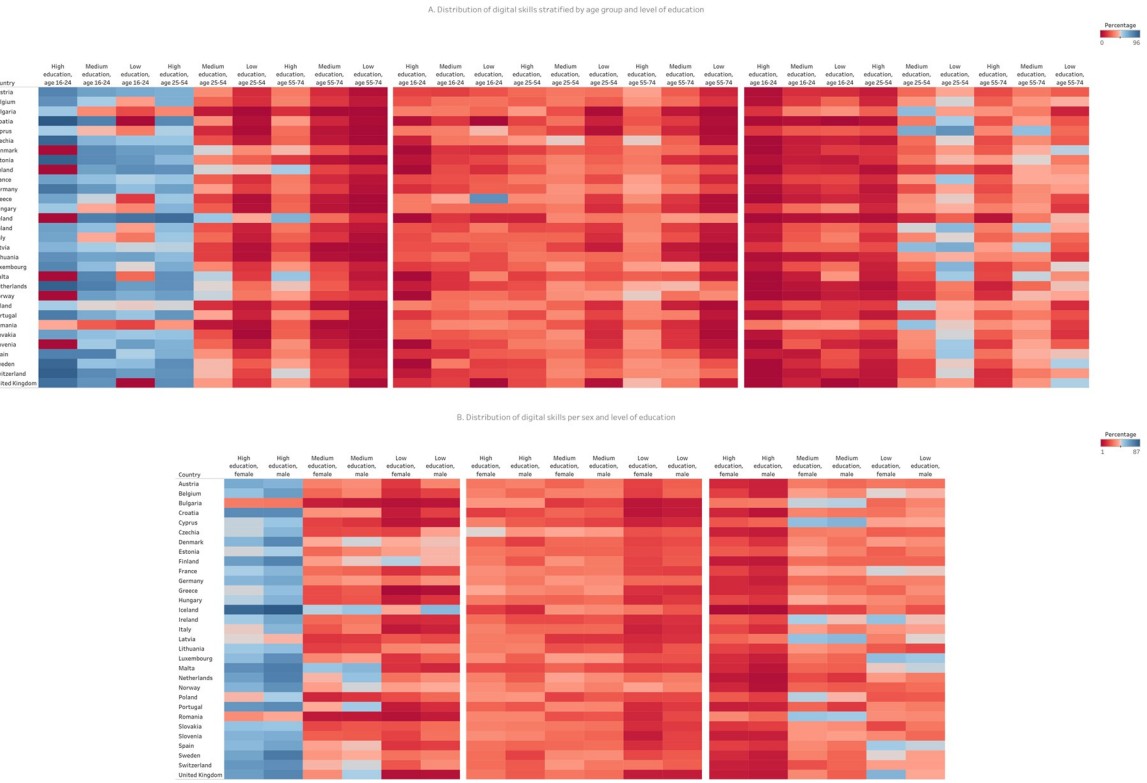

**Fig 5.** Heat map of digital skills with levels of education sub-stratified by age (Fig 5A) and sex (Fig 5B). The left plot indicates above basic digital skills, the middle plot basic digital skills, and the right plot low digital skills.

differences within countries also become evident. People who are younger, higher educated, male, live in urban regions, are either a student or employed, or are employed consistently report higher internet access and digital skills compared to other demographic groups. While previously reported statistics for the European region have shown high internet access [20,21,28], these numbers can be misleading considering the intra- and extra-group demographic distributions of European countries. Findings in terms of overall Internet access are comparable to previous research (88% versus 89.12% [28]), yet geographical discrepancies are glaring (75-98% access rate). Notably - even though Internet pricing is lower and they have a higher proportion of economically active people (see Fig A in S1 Text) - the development of digital skills continues to lag behind in South-Eastern Europe.

According to the directed acyclic graph in Fig 1, a plethora of factors has to work together in order for high levels of digital skills to develop. However, the reported spread of above basic digital skills may indicate that these factors work together inconsistently or insufficiently. With the 2019 Digital Competence Framework now calling for more advanced digital competencies [33], this becomes especially alarming considering only 34.65%(12.13%) of respondents in 2019 had above basic skills under the 2016 classification and these skills are predominantly found in specific population groups (young people, people with high income, employed people and students, and people living in urban areas).

Next to the individual factors, there are also system-level factors that need to be considered, even though they have not been mapped by this particular dataset. As per Greenhalgh's theory on the diffusion of innovation [23], it is important to consider whether – at a system level – there is a need and desire to innovate, whether countries possess the available resources

and the necessary leadership to instigate and sustain change, and whether countries have a politico-cultural climate and population structure that facilitates the uptake of digital technologies. This same rationale applies when examining the uptake of digital technologies across demographic groups. For instance, younger people, students, and urban people are more likely to be raised in a digitally-enabled environment [22,34,35], which may contribute to easier assimilation of digital technologies and, in turn, result in greater digital skills development. At the same time, the groups that could potentially benefit most from the uptake of digital technologies tend to be the groups that are faced with the biggest barriers to access [6,12,13], whether that be due to socioeconomic status, age, health status, or other factors.

Finding and sustaining employment can also become more difficult with increasing digitalisation if the digital skills of the current and prospective workforce are not adequately developed [36]. Simultaneously, as healthcare and public health continue to digitalise, access to health services may become jeopardised without sufficiently digital skills [6,13,22,24]. Health may further deteriorate due to poor digital skills if access to healthcare, social support and employment become more contingent on the possession and mastery of digital skills, further exacerbating inequalities. It is therefore not only crucial from an economic perspective to invest in developing the general population's digital literacy and capacity, but also from a public health perspective. This applies even further when considering the strong correlation between capital stock, income, and earnings with digital skills development. As a result, less developed European countries may find themselves in an unfavourable position relative to their developed peers. These countries will have difficulties catching up, while the skill-biased technological change may deteriorate their cross-country competitiveness.

There are some limitations that have to be considered. Even though this article is underpinned by a large, representative data sample, we have to be mindful of differences among national statistical institutes regarding sampling design [27]. Some countries use samples based on individuals as primary sampling units, while others represent primary sampling units as households in the public register. Gender is also not included separately in this survey, so all gender-related items were inferred using sub-categorisations of education level. These findings are particularly relevant for the age range 16-74 and should not be blindly translated to younger or older age groups. Even though the study findings were based on directed acyclic graphs to infer causality [37], this study remains cross-sectional in nature and true causality cannot be determined. Moreover, the way in which digital skills are outlined in the framework may not be fully representative of the required competencies for Europeans to fully benefit from digital transformations in a post-pandemic world. Regardless, in line with previous research, it can be reasonably assumed that these methodological limitations should not undermine the study findings.

## Implications for future research

This study first and foremost highlighted that Europe cannot be considered a monolith when it comes to internet access and digital literacy. While we advocate for country-level comparisons, we believe splitting Europe into five regions may be the most valuable for accurate comparisons in case a study design requires countries to be pooled into clusters (North-West, North-East, Center, South-West, and South East). This ensures countries with similar internet access and digital skill levels are clustered, thus removing the bias they would introduce if these regions were lumped together. Further research into the cultural and motivational factors is also warranted to further the understanding of why certain areas in Europe experience difficulties in connecting to the Internet and developing digital skills. Finally, this article is only a single time point in determining the digital capacities of Europe. Therefore, we

recommend that this article be replicated using newer data once available in order to establish longitudinal trends. Another recommendation would be to overlay this research with global trends across similar demographics.

## Implications and recommendations for policy and governance

The initial economic endowments are a prerequisite for establishing the infrastructure, acquiring the necessary equipment, and developing digital skills. Costly infrastructural ICT investments must be considered public goods to be carried out through public investment or public-private partnerships and under the supervision of central authorities to address existing inequalities. Should the provision of necessary public goods be left to profit-centred, free-market stakeholders, there is a concern that digital technologies will remain under-provided and inequitably distributed across Europe. However, as the eco-system becomes more person-centric, there may be a case for the private sector to become first movers in innovative ICT investments and increasing the digital literacy of prospective consumer groups. This can be encouraged and supported by government by keeping legal and regulatory barriers to entry low. In the case of other public goods (e.g., roads, railways, healthcare facilities), the scenario of inequitable distribution will generate market failure if not adequately addressed. In line with Mazzucato's recommendations, governments must shift away from the neoliberal ideology that they are only tasked to fix and repair and instead, towards a mindset of innovation, catalysing the economy to be more purpose-driven and goal-oriented to achieve sustainable development [38]. This entails the development of digital capacity in-government and across the general population. Following that reasoning, traditional definitions of digital literacy and approaches to building digital capacity must be expanded and updated to adapt to the pace of development brought about by the COVID-19 pandemic. Furthermore, it is crucial for curricula and teaching practice to be updated to better reflect and incorporate this new definition/conceptual framework of digital literacy. However, in doing so, it is paramount that digitalisation – particularly in education practice – is used as a means to an end, not a solution in and of its own [11,22].

Acknowledgement of these conclusions under the framework of European (in particular EU) institutional design brings about an additional dimension of complexity. Given the vast differences in available resources and absence of the supranational political climate delivering inclusive strategies, digital divides threaten to become principal contemporary forces leading the cross-country divergence. Coupled with the core-periphery divide, digital divides impede the European objectives of promoting all citizens' well-being, combatting social exclusion and inequality, reducing developmental disparities and the backwardness of the least-favoured regions, and ensuring harmonized socio-economic growth. Hence, to prevent a growing gap and halt the continuous suffering of less-developed areas, the provision of necessary public goods cannot be left in the private nor country-level sphere alone and must be addressed at a supranational level. European policymakers must prioritize equitable and sustainable development and make better use of existing facilities financed through direct transactions and delivered through a combination of private and public arrangements. Aligned with this agenda, a step in the right direction would be to approve the NextGenerationEU instrument. This recovery and transformative agreement aims to target digital transition (Digital Europe Program) and education and training to support digital skills development at the European level.

## Conclusion

The transition into the Digital Age provides novel opportunities to minimise existing inequalities and pursue the Agenda for Sustainable Development. However, to effectively pursue this

goal, we must first ensure that inequalities are not exacerbated during this transition. Based on the current findings, we conclude that Europe's digital climate at the start of the pandemic left much to be desired, showcasing large in- and cross-country discrepancies in internet access and digital skills. Although the pandemic forced society to digitalise, it remains to be seen how much sticks around. One conclusion, however, is certain: Europe cannot be considered a homogenous zone in terms of internet access or digital skills and pursuing a complete digital transition would currently exacerbate in- and cross-country inequalities.

## Supporting information

**S1 Text. Supplementary Material for Online Publication.**
(DOCX)

## Author Contributions

**Conceptualization:** Robin van Kessel, Brian Li Han Wong, Ivan Rubinić, Katarzyna Czabanowska.

**Data curation:** Robin van Kessel, Brian Li Han Wong, Ivan Rubinić.

**Formal analysis:** Robin van Kessel, Brian Li Han Wong, Ivan Rubinić, Katarzyna Czabanowska.

**Investigation:** Robin van Kessel, Brian Li Han Wong, Ivan Rubinić.

**Methodology:** Robin van Kessel, Brian Li Han Wong, Ivan Rubinić.

**Project administration:** Robin van Kessel.

**Resources:** Robin van Kessel.

**Supervision:** Robin van Kessel, Katarzyna Czabanowska.

**Validation:** Robin van Kessel, Ivan Rubinić, Ella O'Nuallain.

**Visualization:** Robin van Kessel, Ivan Rubinić.

**Writing – original draft:** Robin van Kessel, Brian Li Han Wong, Ivan Rubinić, Katarzyna Czabanowska.

**Writing – review & editing:** Robin van Kessel, Brian Li Han Wong, Ivan Rubinić, Ella O'Nuallain, Katarzyna Czabanowska.

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
