## [Decision Letter · Decision Letter 0]

26 Oct 2021

PDIG-D-21-00009

Is Europe prepared to go digital? Making the case for developing digital capacity: an exploratory analysis of Eurostat survey data

PLOS Digital Health

Dear Dr. Wong,

Thank you for submitting your manuscript to PLOS Digital Health. After careful consideration, we feel that it has merit but does not fully meet PLOS Digital Health’s publication criteria as it currently stands. Therefore, we invite you to submit a revised version of the manuscript that addresses the points raised during the review process.

In addition to the reviewers comments please pay attention to the comments below from the editor:

The paper is fairly well-written. The recommendations are strong with major policy and practice implications in advocating for digital transformation in Europe for better population health and wellbeing.  However, the results are not so clear which raise issues about the discussion.

Suggestions for improvement: The authors need to revisit their analysis and also go through the entire manuscript and abstract to edit for language.  There is need for consistency in the use of Internet as in some places you use the lowercase. Unless it is being used as a noun and/or appearing at the beginning of a sentence, it ought to be a lowercase.

Needs editing for language. Here are some examples

The sentence,  ‘Previous studies indicated that Europe has an average Internet access rate of 90%, yet rarely specify for different demographics and does not report on the presence of digital skills’. Since you are referring to plural, does needs to be do

We look forward to receiving your revised manuscript.

Kind regards,

Pauline Bakibinga, M.D, Ph.D

Guest Editor

PLOS Digital Health

Journal Requirements:

1. Please update the completed 'Competing Interests' statement, including any COIs declared by your co-authors. If you have no competing interests to declare, please state "The authors have declared that no competing interests exist". Otherwise please declare all competing interests beginning with the statement "I have read the journal's policy and the authors of this manuscript have the following competing interests:"

3. Please note that your Data Availability Statement is currently missing a direct link to access each database. If your manuscript is accepted for publication, you will be asked to provide these details on a very short timeline. We therefore suggest that you provide this information now, though we will not hold up the peer review process if you are unable.

4. Please provide separate figure files in .tif or .eps format only and remove any figures embedded in your manuscript file. Please ensure that all files are under our size limit of 20MB.  

Once you've converted your files to .tif or .eps, please also make sure that your figures meet our format requirements

For more information about how to convert your figure files please see our guidelines: https://journals.plos.org/digitalhealth/s/figures

5. We have noticed that you have uploaded supporting information but you have not included a list of legends. Please add a full list of legends for all supporting information files (including figures, table and data files) after the references list.

6. Please provide us with a direct link to the base layer of the map used in Figures 2, 3, and supplementary eFigure 1, and ensure this location is also included in the figure legend. 

Please note that, because all PLOS articles are published under a CC BY license (creativecommons.org/licenses/by/4.0/), we cannot publish proprietary maps such as Google Maps, Mapquest or other copyrighted maps. If your map was obtained from a copyrighted source please amend the figure so that the base map used is from an openly available source.

Please note that only the following CC BY licences are compatible with PLOS licence: CC BY 4.0, CC BY 2.0  and CC BY 3.0, meanwhile such licences as CC BY-ND 3.0 and others are not compatible due to additional restrictions. If you are unsure whether you can use a map or not, please do reach out and we will be able to help you. 

The following websites are good examples of where you can source open access or public domain maps:

Additional Editor Comments (if provided):

Reviewers' comments:

Reviewer's Responses to Questions

**Comments to the Author**

1. Does this manuscript meet PLOS Digital Health’s publication criteria? Is the manuscript technically sound, and do the data support the conclusions? The manuscript must describe methodologically and ethically rigorous research with conclusions that are appropriately drawn based on the data presented.

Reviewer #1: Yes

Reviewer #2: Yes

Reviewer #3: Yes

2. Has the statistical analysis been performed appropriately and rigorously?

Reviewer #1: Yes

Reviewer #2: Yes

Reviewer #3: Yes

3. Have the authors made all data underlying the findings in their manuscript fully available (please refer to the Data Availability Statement at the start of the manuscript PDF file)?

Reviewer #1: Yes

Reviewer #2: Yes

Reviewer #3: Yes

4. Is the manuscript presented in an intelligible fashion and written in standard English?

Reviewer #1: Yes

Reviewer #2: Yes

Reviewer #3: Yes

5. Review Comments to the Author

Reviewer #1: I have reviewed this informative study. It is a matter of great pleasure to assess this informative study. I believe the authors have examined a good research topic entitled, “Is Europe prepared to go digital? Making the case for developing digital capacity: an exploratory analysis of Eurostat survey data.”

In my assessment, the writing of this article looks sounds good with a creative research topic. I have some suggestions to the authors to improve the quality of this article. Overall, it a good article that offers useful insight for the scholars.

I want to accept your study for publication after minor changes as suggested. Before endorsing your study for publication, you need to work on my suggestions. The suggested articles are published in leading SSCI journals. By following these studies, your article will be improved.

Abstract of the study

The abstract is proper and shows good connection of the study. I advise the authors to recheck the abstract and fix minor grammar errors. The abstract must reflect high quality, as it the "FACE" of the study.

English level

I suggest the authors to focusing on checking some typo errors to make it easy to understand for the readership of the journal.

Introduction section

I strongly advise the authors improve introduction according to suggested articles in the introduction section. These research articles have identified health-related topics. I believe it will improve the quality of your work. I strongly suggested them to improve this section a bit more. I advise authors to revisit their literature section of the recommended studies and cite these studies to enhance your research study's quality to reach scientific merit for publication.

Azizi, M. R., Atlasi, R., Ziapour, A., & Naemi, R. (2021). Innovative human resource management strategies during the COVID-19 pandemic: A systematic narrative review approach. Heliyon, 6(12).

Abbas, J., Aqeel, M., Jaffar, A., Nurunnabi, M., and Bano, S. (2019). "Tinnitus perception mediates the relationship between physiological and psychological problems among patients." Journal of Experimental Psychopathology, 10(3), 2043808719858559.

Aqeel, M., et al., The Influence of Illness Perception, Anxiety and Depression Disorders on Students Mental Health during COVID-19 Outbreak in Pakistan: A Web-Based Cross-Sectional Survey. International Journal of Human Rights in Healthcare, 2020. 14.

Abbas, J., Wang, D., Su, Z., & Ziapour, A. (2021). The Role of Social Media in the Advent of COVID-19 Pandemic: Crisis Management, Mental Health Challenges and Implications. Risk Manag Healthc Policy, Volume 14, 1917-1932. doi:10.2147/rmhp.S284313

Su, Z., McDonnell, D., Wen, J., Kozak, M., Šegalo, S., . . . Xiang, Y.-T. (2021). Mental health consequences of COVID-19 media coverage: the need for effective crisis communication practices. Globalization and Health, 17(1), 4. doi:10.1186/s12992-020-00654-4

Literature sections

I recommend the authors add suggested articles in the literature section. These research articles have identified health-related topics. I believe it will improve the quality of your work. I advise authors to revisit their literature section of the recommended studies to enhance your research study's quality to reach scientific merit for publication.

I want to see publish this creative work after some corrections. I have endorsed this study as; it deserves the merit for publication. However, I suggest the authors make minor corrections according to my advice. The authors add the latest citations about infectious disease. Please read the suggested studies and execute them in the introduction, literature, and method sections. How social media and internet use among students is helpful. Add few lines in the introduction and literature sections.

NeJhaddadgar, N., Ziapour, A., Zakkipour, G., Abolfathi, M., & Shabani, M. (2020, 2020/11/13). Effectiveness of telephone-based screening and triage during COVID-19 outbreak in the promoted primary healthcare system: a case study in Ardabil province, Iran. Journal of Public Health. https://doi.org/10.1007/s10389-020-01407-8

Abbas, J. (2021, 2021/02/23/). Crisis management, transnational healthcare challenges and opportunities: The intersection of COVID-19 pandemic and global mental health. Research in Globalization, 100037. https://doi.org/10.1016/j.resglo.2021.100037

Maqsood, A., Abbas, J., Rehman, G., & Mubeen, R. (2021, 2021/11/01/). The paradigm shift for educational system continuance in the advent of COVID-19 pandemic: Mental health challenges and reflections. Current Research in Behavioral Sciences, 2, 100011. https://doi.org/10.1016/j.crbeha.2020.100011

Shuja, K. H., Aqeel, M., Jaffar, A., & Ahmed, A. (2020, Spring). COVID-19 Pandemic and Impending Global Mental Health Implications. Psychiatr Danub, 32(1), 32-35. https://doi.org/10.24869/psyd.2020.32

Su, Z., Wen, J., Abbas, J., McDonnell, D., Cheshmehzangi, A., Li, X., Ahmad, J., Segalo, S., Maestro, D., & Cai, Y. (2020, Dec). A race for a better understanding of COVID-19 vaccine non-adopters. Brain Behav Immun Health, 9, 100159. https://doi.org/10.1016/j.bbih.2020.100159

Materials and Methods

This section indicates how you arranged your article. You can see the suggested study and improve your method section.

Abbas, J., Aqeel, M., Abbas, J., Shaher, B., A, J., Sundas, J., and Zhang, W. (2019). "The moderating role of social support for marital adjustment, depression, anxiety, and stress: Evidence from Pakistani working and nonworking women." J Affect Disord, 244, 231-238.

Local Burden of Disease, H. I. V. C. (2021). Mapping subnational HIV mortality in six Latin American countries with incomplete vital registration systems. BMC Medicine, 19(1), 4. doi:10.1186/s12916-020-01876-4

Abbas, J., Aman, J., Nurunnabi, M., & Bano, S. (2019). The Impact of Social Media Learning Behavior for Sustainable Education: Evidence of Students from Selected Universities in Pakistan. Sustainability, 11(6), 1683.

Abbasi, K. R., Abbas, J., and Tufail, M. (2021). "Revisiting electricity consumption, price, and real GDP: A modified sectoral level analysis from Pakistan." Energy Policy, 149, 112087.

Yoosefi Lebni, J., Abbas, J., Khorami, F., Khosravi, B., Jalali, A., and Ziapour, A. (2020). "Challenges Facing Women Survivors of Self-Immolation in the Kurdish Regions of Iran: A Qualitative Study." Frontiers in psychiatry, 11, 778.

Results

The results section looks good. The authors can refine it by removing some typo errors.

Discussion section

Briefly discuss the contribution to the scientific literature. Add few lines on contribution of this study how results are insightful for academic purpose. Improve this section. Please see suggested studies in this section.

Su, Z., McDonnell, D., Cheshmehzangi, A., Abbas, J., Li, X., & Cai, Y. (2021). The promise and perils of Unit 731 data to advance COVID-19 research. BMJ Global Health, 6(5), e004772. https://doi.org/10.1136/bmjgh-2020-004772

Abbas, J. (2020, Autumn - Winter). The Impact of Coronavirus (SARS-CoV2) Epidemic on Individuals Mental Health: The Protective Measures of Pakistan in Managing and Sustaining Transmissible Disease. Psychiatr Danub, 32(3-4), 472-477. https://doi.org/10.24869/psyd.2020.472

Yoosefi Lebni, J., Abbas, J., Moradi, F., Salahshoor, M. R., Chaboksavar, F., Irandoost, S. F., Nezhaddadgar, N., & Ziapour, A. (2020, Jul 2). How the COVID-19 pandemic effected economic, social, political, and cultural factors: A lesson from Iran. International Journal of Social Psychiatry, 20764020939984. https://doi.org/10.1177/0020764020939984

Abbas, J., Mubeen, R., Iorember, P. T., Raza, S., & Mamirkulova, G. (2021). Exploring the impact of COVID-19 on tourism: transformational potential and implications for a sustainable recovery of the travel and leisure industry. Current Research in Behavioral Sciences, 2, 100033.

Implications

Add this separate heading and discuss implications

Conclusion

Make a separate heading for this section. It should present a good picture of the study. I want to see this manuscript published as it has presented a good research topic, although it needs minor corrections, which can be fixed in the revised version. Pay attention of English quality to reach scientific merit. I accept and endorse this manuscript for publication after the suggested minor corrections.

Figures & Tables

I suggest to add some figures and more tables in this study

Reviewer #2: The data are well presented and support the discussions/conclusions as well as recommendations for policy and governance. The tables as well as figures included in the paper are clear and easy to understand. The analysis clearly shows that there are demographic and other factors affecting access to and capacities in use of digital tools and applications.

Reviewer #3: p.4 The Community Survey and Data Access

This paragraph is too short and lack info on important characteristics of the survey:

- How were households selected?

- The sentence "the survey data [...] represents approximately 75% of the population between the ages of 16

and 74" is ambiguous. A sample size of about 200000 individuals is by all means a large sample size.

p.4 First paragraph in "Variable Selection" is unclear

p.4 Sentence "Sub-stratification of education levels was possible by means of sex and three secondary age groups (16-24, 25-54, and 55-74

years)." is unclear.

p.5 Info such as "Data was cleaned in R version 4.0.2, specifically using the ‘dplyr’ and ‘reshape2’ packages" is irrelevant. Provide the R code through a github repository instead.

p.5 Fig.1 does not make much sense without any explanation

p.5 Descriptive Statistics: Given the sample size, I am puzzled by the high values of the standard deviations reported.

As a quick approximation, for a simple random survey, the classical formula V(p) = (1-n/N)x(N/(N-1))xp(1-p)/n leads to estimated SD's several orders of magnitude smaller than those reported. Is there the possibility of a numerical mistake?

p.6 "Most individual respondents had above basic digital skills (34.65%)"

People tend to use "most" to mean anything over 50%

Table 1 would be more efficiently placen in SM and replaced in the main text by a couple of figures.

The whole data analysis reporting is labored and not easy to read. It should be better streamlined with an effort to focus on a few key results.

I found the discussion rather speculative, a bit verbose and somehow disconnected from the data analysis.

The final sentence "Europe is still unfit to sustainably host a digital society and doing so would exacerbate in- and cross-country inequalities" does not seem well motivated by the main findings and unduly worrisome.

6. PLOS authors have the option to publish the peer review history of their article (what does this mean?). If published, this will include your full peer review and any attached files.

**Do you want your identity to be public for this peer review?** For information about this choice, including consent withdrawal, please see our Privacy Policy.

Reviewer #1: No

Reviewer #2: **Yes: **Neeraj Kak

Reviewer #3: No

---

## [Decision Letter · Decision Letter 1]

10 Dec 2021

Is Europe prepared to go digital? Making the case for developing digital capacity: An exploratory analysis of Eurostat survey data

PDIG-D-21-00009R1

Dear Dr. Wong,

We're pleased to inform you that your manuscript has been judged scientifically suitable for publication and will be formally accepted for publication once it meets all outstanding technical requirements. 

Within one week, you'll receive an e-mail detailing the required amendments. When these have been addressed, you'll receive a formal acceptance letter and your manuscript will be scheduled for publication. The journal will begin publishing content in early 2022.

An invoice for payment will follow shortly after the formal acceptance. To ensure an efficient process, please log into Editorial Manager at https://www.editorialmanager.com/pdig/ click the 'Update My Information' link at the top of the page, and double check that your user information is up-to-date. If you have any billing related questions, please contact our Author Billing department directly at authorbilling@plos.org.

Kind regards,

Laura Sbaffi, PhD, MA, MSc

Section Editor

PLOS Digital Health

Additional Editor Comments (optional):

Reviewers' comments:

Reviewer's Responses to Questions

**Comments to the Author**

1. If the authors have adequately addressed your comments raised in a previous round of review and you feel that this manuscript is now acceptable for publication, you may indicate that here to bypass the “Comments to the Author” section, enter your conflict of interest statement in the “Confidential to Editor” section, and submit your "Accept" recommendation.

Reviewer #1: All comments have been addressed

Reviewer #2: All comments have been addressed

Reviewer #3: All comments have been addressed

2. Does this manuscript meet PLOS Digital Health’s publication criteria? Is the manuscript technically sound, and do the data support the conclusions? The manuscript must describe methodologically and ethically rigorous research with conclusions that are appropriately drawn based on the data presented.

Reviewer #1: Yes

Reviewer #2: Yes

Reviewer #3: Yes

3. Has the statistical analysis been performed appropriately and rigorously?

Reviewer #1: Yes

Reviewer #2: Yes

Reviewer #3: Yes

4. Have the authors made all data underlying the findings in their manuscript fully available (please refer to the Data Availability Statement at the start of the manuscript PDF file)?

Reviewer #1: Yes

Reviewer #2: Yes

Reviewer #3: Yes

5. Is the manuscript presented in an intelligible fashion and written in standard English?

Reviewer #1: (No Response)

Reviewer #2: Yes

Reviewer #3: Yes

6. Review Comments to the Author

Reviewer #1: The paper can be accepted.

Reviewer #2: The authors have responded to the comments from reviewers.

Reviewer #3: I am not a big fan of the third review round but I would encourage you to provide more info on the calculation of the standard deviations somewhere as SM

7. PLOS authors have the option to publish the peer review history of their article (what does this mean?). If published, this will include your full peer review and any attached files.

**Do you want your identity to be public for this peer review?** For information about this choice, including consent withdrawal, please see our Privacy Policy.

Reviewer #1: No

Reviewer #2: **Yes: **Neeraj Kak

Reviewer #3: **Yes: **Gilles B. Guillot
